# From Sub-Pectoral to Pre-Pectoral Implant Reconstruction: A Decisional Algorithm to Optimise Outcomes of Breast Replacement Surgery

**DOI:** 10.3390/healthcare11050671

**Published:** 2023-02-24

**Authors:** Glenda Giorgia Caputo, Sebastiano Mura, Filippo Contessi Negrini, Roberta Albanese, Pier Camillo Parodi

**Affiliations:** Plastic and Reconstructive Surgery, DAME—Department of Medical Area, University Hospital of Udine, 33100 Udine, Italy

**Keywords:** breast implant replacement surgery, secondary breast implant surgery, pocket exchange, pre-pectoral breast implant, animation deformity

## Abstract

Background: Innovations and advancements with implant-based breast reconstruction, such as the use of ADMs, fat grafting, NSMs, and better implants, have enabled surgeons to now place breast implants in the pre-pectoral space rather than under the pectoralis major muscle. Breast implant replacement surgery in post-mastectomy patients, with pocket conversion from retro-pectoral to pre-pectoral, is becoming increasingly common, in order to solve the drawbacks of retro-pectoral implant positioning (animation deformity, chronic pain, and poor implant positioning). Materials and Methods: A multicentric retrospective study was conducted, considering all patients previously submitted to implant-based post-mastectomy breast reconstruction who underwent a breast implant replacement with pocket conversion procedure at the University Hospital of Udine—Plastic and Reconstructive Surgery Department—and “Centro di Riferimento Oncologico” (C.R.O.) of Aviano, from January 2020 to September 2021. Patients were candidates for a breast implant replacement with pocket conversion procedure if they met the following inclusion criteria: they underwent a previous implant-based post-mastectomy breast reconstruction and developed animation deformity, chronic pain, severe capsular contracture, or implant malposition. Patient data included age, body mass index (BMI), comorbidities, smoking status, pre- or post-mastectomy radiotherapy (RT), tumour classification, type of mastectomy, previous or ancillary procedures (lipofilling), type and volume of implant used, type of ADM, and post-operative complications (breast infection, implant exposure and malposition, haematoma, or seroma). Results: A total of 31 breasts (30 patients) were included in this analysis. Just three months after surgery, we recorded 100% resolution of the problems for which pocket conversion was indicated, which was confirmed at 6, 9, and 12 months post-operative. We also developed an algorithm describing the correct steps for successful breast-implant pocket conversion. Conclusion: Our results, although only early experience, are very encouraging. We realized that, besides gentle surgical handling, one of the most important factors in proper pocket conversion selection is an accurate pre-operative and intra-operative clinical evaluation of the tissue thickness in all breast quadrants.

## 1. Introduction 

Breast reconstruction surgery is one of the wider fields in which plastic surgeons work. In fact, breast reconstruction takes into consideration both the autologous reconstruction technique by free or pedicle flap, the prosthetic reconstruction in all its variance, or a combination of these two ambits in the hybrid reconstruction, where prosthesis and autologous tissue such as adipose tissue graft are combined to achieve the best possible result. Perhaps, more than in other branches of plastic surgery, breast reconstruction has to aim for the most harmonic and aesthetic result due to the symbolic role of the breast as a self-defining characteristic and because of its sexual involvement, which represents one of the most difficult challenges in this field. At the very beginning of this sub-speciality, breast volume was the main target of breast reconstruction; nowadays, surgeons and patients judge the final reconstruction results on every single detail: breast volume, breast ptosis, nipple position, symmetry of the reconstructed breast compared with the contralateral one and the harmonic result considering the whole patient physique. This strong scope has led plastic surgeons to look for any technical and technological improvement to reach the best outcome. The history of breast reconstruction is studded with numerous technical variants, dictated both from surgeon intuition and ability as well as the available technology. Nowadays, innovations and advancements in prosthetic breast reconstruction, thanks to the progress in bio-engineering and bio-materials, such the introduction of the acellular dermal matrix (ADM), the acquired knowledge on fat grafting, and the development of better implants, associated with a new conservative technique of nipple sparing mastectomy (NSM), have enabled surgeons to now place breast implants in the pre-pectoral space rather than under the pectoralis major muscle [1,2,3,4]. In the past, different attempts to place the implant in the subcutaneous area have been made. In fact, this is the natural place to put a breast implant; as a matter of fact, it is the anatomical glandular tissue natural position. Pre-pectoral reconstruction was initially proposed in the 1970s, but was quickly abandoned because of the high complication rates, such as implant or reconstruction failure [5]. Failures of the reconstructions were associated with a high rate of implant exposure due to the thinning out of the mastectomy flaps and the consequent wound dehiscences and infection of the tissues. Thanks to the introduction of ADM and conservative mastectomies, this technique was reconsidered: the purpose of the acellular dermal matrix and, in part, of the synthetic meshes is to reinforce the breast skin. The meshes are projected to integrate to the subcutaneous tissue of the mastectomy flaps, so as to thicken it and give it more structure and more vascularised tissue, stimulating tissue regeneration. Furthermore, ADMs help the mastectomy flap to bear the loads of the prosthesis and to create an interface between the implant and the overlying skin, as it can be anchored to a deeper tissue layer such as the pectoralis fascia. The need to not elevate the pectorals major is not limit to the like-with-like reconstruction *per se*, but it is demonstrated that the advantages of pre-pectoral implant placement include a reduction in post-operative pain, the absence of animation deformity (AD), and the possibility to recreate a more natural breast shape with more age-appropriate ptosis, without the upper dislocation of the breast implant due to the muscular retraction. Nowadays, this newly accumulated experience in the breast reconstruction field and the availability of the above mentioned new technologies have set a new target in breast surgery: the correction of the side effects presented by previous reconstruction modalities. In particular, we had to face patients complaining about sub-muscular breast implant positioning drawbacks, so the pocket exchange technique was introduced. Pocket exchange consists of a revision implant procedure where the surgeon creates a new implant pocket by dissecting the virtual plane between the pectoralis major and the overlying subcutaneous tissue, then removes the previous implant and insert the new prosthesis in the newly created pre-pectoral pocket. There are different possibilities to manage the muscle and the previous implant capsule: someone anchor the pectoralis major to the costal plane or to the posterior capsule layer; someone prefer to always perform total capsulectomy, others prefer to be more conservative, performing only capsulotomies if necessary. The concept is to convert the reconstruction from retro- to pre-muscular. The functional and cosmetic benefits of implant pocket conversion from retro-pectoral to pre-pectoral are well described in the literature; however, no article has yet tackled the decision-making process underlying it. The primary indications for this procedure are: (i) Animation deformity (AD) created by the pectoralis major activation, which moves the prosthesis cranially with each muscle contraction. Pocket exchange has demonstrated a 100% success rate [6,7,8]. (ii) Chronic pain (CP), because the pre-pectoral implant position is less painful than the submuscular one [9]. (iii) Poor implant positioning, e.g., asymmetry due to lateral or cranial malposition accompanied with a decrease in breast ptosis. (iv) Finally, capsular contracture (CC) [10]. In our practice, following patient wishes to resolve one or more of the above-mentioned submuscolar reconstruction stigma, we increasingly perform implant pocket conversion from retro-pectoral to pre-pectoral during breast implant replacement surgery in post-mastectomy patients, achieving a more natural result [11]. However, this procedure cannot be performed in all cases, especially if the skin envelope does not have a satisfactory thickness and consistency. In fact, such as in the pre-pectoral direct-to-implant breast reconstruction, there are some relative contraindications in the pocket exchange manoeuvre: in our opinion, patients who present poor mastectomy skin quality as a result of irradiated breast or thin mastectomy flaps, have to be prepared for the pocket exchange with at least one lipofilling. We propose here an in-house algorithm, based on a retrospective analysis of our clinical experience, suggesting a step-by-step decision-making process, in order to standardize case selection and ensuring successful breast implant pocket conversion. 

## 2. Materials and Methods

After approval from the institutional review committee, we set up a multicentric retrospective study involving all patients previously submitted to implant-based post-mastectomy breast reconstruction who underwent a breast implant replacement with pocket conversion procedure at the University Hospital of Udine—Plastic and Reconstructive Surgery Department—and “Centro di Riferimento Oncologico” (C.R.O.) of Aviano, from January 2020 to September 2021. These patients were identified and provided informed consent for the use of their clinical data. Patients were candidates for a breast implant replacement with pocket conversion procedure if they met the following inclusion criteria: they underwent a previous implant-based post-mastectomy breast reconstruction and developed animation deformity, chronic pain, severe capsular contracture, or implant malposition with a marked asymmetry. Patient data included age, body mass index (BMI), comorbidities, smoking status, pre- or post-mastectomy radiotherapy (RT), tumour classification, type of mastectomy, previous or ancillary procedures (lipofilling), type and volume of implant used, type of ADM, and post-operative complications (breast infection, implant exposure and malposition, haematoma, or seroma). In the presence of one of the previously cited situations (animation deformity, chronic pain, severe capsular contracture, or implant malposition) clinical assessment was conducted first so as to appraise cutaneous envelope thickness and implant capsule status. If, at the pre-operative evaluation, the cutaneous envelope thickness was judged to be adequate (pinch test > 2 cm), as well as during the intraoperative evaluation, pocket exchange was performed. Vice versa, in the presence of a poor thickness for the upper pole, the lower pole, or both, a preliminary session of lipofilling was performed, and the patient was clinically reassessed three months after. Concerning the management of the implant capsule, we opted for a sub-total capsulectomy (limited to the anterior sheet) in cases of high-grade capsular contracture (Becker III or IV), and for a partial capsulectomy (mainly in Becker III) or inferior pole capsulotomy (mainly in Becker II) to adjust the implant pocket for the new implant. When unnecessary, we preferred to avoid sub-pectorals capsulectomy to reduce the ematoma formation risk. During surgery, we first reached the pre-capsular plane by making the incision through the previous mastectomy scar, which was more often radial between the outer quadrants. We then followed the pre-pectoral plane upwards to create the new pre-pectoral implant pocket: in this phase, it is of the utmost importance to preserve all of the subcutaneous tissue in order to maintain the mastectomy flap thickness. In capsulectomy cases, we followed the pre-capsular plane down into the lower pole, detaching the fat tissue from the capsule as we reached the infra-mammary fold (IMF); while in the capsulotomy cases, we directly cut the capsule together with the skin to access the periprosthetic pocket and eventually make capsulotomies. Meanwhile, in the upper pole, if the capsule was particularly hard, we accurately detached it as little as possible from the pectoralis major, to minimize damage to the muscle, and leave it on the back surface of the pectoralis major. Once the pectoralis major was freed, we normally anchored the muscle to the chest wall (costal periosteum or posterior capsular layer) via resorbable sutures (Vicryl 2/0) and fixed one drain in the new pre-pectoral pocket. Remodelling the implant pocket was another crucial point in this surgery to achieve the best result: in our practice, when the capsule was conserved, we frequently performed some radial or circumferential capsulotomy at the lower pole to extend it and to reach a more natural ptosis degree, always taking attention to obtain the highest degree of symmetry with the contralateral breast. Considering the upper pole, the level of dissection of the pectoralis major from the overlying skin defined the new upper pocket limit. We then chose the new implant, placing implant sizers with the patient in a sitting position and the arms adducted. Once the correct volume and form of the implant was decided, we wrapped it up entirely with ADM (Braxon^®^), and then either fixed it to the chest wall with four or five stiches, or directly positioned the implant in the new pocket, closing the skin using layered sutures. The decision to wrap or not the microtextured implant with an ADM was taken intraopertively based on the thickness of the skin flap and the necessity to perform a total capsulectomy or not: we wrapped the implant in case of anterior capsule layer removal. The clinical results, in terms of the resolution of animation deformity, chronic pain, severe capsular contracture, or implant malposition, were assessed at 3 months after surgery. Moreover, a clinical assessment was conducted 6, 9, and 12 months after pocket conversion surgery. It is important to note that four patients underwent one session of lipofilling, and one underwent five sessions (for a total of nine sessions in five patients), before the procedure of implant exchange and pocket conversion. They received an average of 108.3 cc (range 75–150 cc) per session, distributed in all quadrants of the breast in the subcutaneous layer. The fat was harvested from the trochanteric region (five cases), or the hips and upper abdomen (four cases) after local infiltration of the Klein solution. The fat was left to settle until the liquid part separated from the dense part, and the fat graft was infiltrated through 1.6 mm diameter cannulas with 2.5 cc syringes. The lipograft was performed on average nine months after the mastectomy (range six to twelve months). The patient who had five sessions of lipofilling, after the first one, which was performed after six months from the mastectomy, received the remaining four lipofilling 6, 9, 12, and 18 months from the first one.

## 3. Results

A total of 31 breasts (30 patients) were included in this analysis: 7 were treated at CRO of Aviano and 24 were treated at the University Hospital of Udine. Patients’ characteristics are reported in Table 1, as well as primary and secondary breast implant characteristics and grade of capsular contracture. 

Indications for the pocket conversion technique were animation deformity in 26 patients (87%), chronic pain in 13 patients (43%), capsular contracture in 17 patients (57%), and breast asymmetry at rest in 19 patients (63%). In our experience, retro-pectoral breast reconstruction drawbacks often overlap in such cases: in our case series, all cases of capsular contracture (17 patients) also had breast asymmetry, the 26 patients with animation deformity overlapped with 11 patients with chronic pain; only three patients had all four drawbacks at the same time and only four patients had one indication for surgery, which was animation deformity; 10 patients had 3 indications and 13 had 2 indications. 

During surgery, in 13 breasts (42%), lower pole capsulectomy was performed; subtotal capsulectomy was performed in only six breasts (19%). In 12 cases (39%), the capsule was preserved and the pocket was rearranged by capsulotomy. Post-procedure complications were observed in three patients (all three had one or more risk factors): there were two cases of wound dehiscence, corrected via scar revision, and one case of hematoma. One patient required a reintervention for the correction of a certain degree of implant malposition. No implants were lost. At three months after surgery, we recorded 100% resolution of the problems for which pocket conversion was indicated, which was confirmed at the following visits at 6, 9, and 12 months (Figure 1 and Figure 2).

## 4. Discussion

Nowadays, one of the most difficult challenges plastic surgeons face is the side effects of previous reconstruction techniques. In breast reconstruction, this problem can be found in patients who received a retro-pectoral reconstruction in the past, showing classical stigma of this reconstruction: animation deformity, chronic pain, or implant malposition such as cranialization. This challenge comes at a time when this complication can be bypassed thanks to new experience and tools. In fact, innovations and advancements with implant-based breast reconstruction are many, and include the use of ADM, fat grafting, nipple sparing mastectomy, and better quality implants: with an anatomical shape, cohesive gel, and micro- or nano-textured shields [12,13]. These advancements have expanded the possibility to achieve better results in breast reconstruction, limiting the post-mastectomy physical and psychological burden to patients. Firstly, these new technologies have enabled surgeons to now place breast implants in the pre-pectoral space, rather than under the pectoralis major muscle, reconsidering a technique that was lost in the past due to the high rate of reconstruction failure. Nowadays, this reconstruction method shows a lower rate of complications than in past experiences: surgeons are well-trained on this technique and they now know how to handle it. Based on the accumulated experience for the pre-pectoral direct to implant breast reconstruction, surgeons have decided to introduce the concept of the pocket exchange during prosthesis revision. In fact, despite the benefits of subpectoral device placement, shortcomings such as animation deformity with muscle contraction, pectoralis muscle spasm, and a generalized discomfort are common, and patients frequently complain about it. In the last few years, following the enthusiasm for the pre-pectoral reconstruction and its promising results, the implant pocket exchange is becoming a popular means to resolve poor outcomes of retro-pectoral IBBR [6,7,14,15]. The primary indications for the implant exchange and pocket conversion procedure are: animation deformity, chronic pain, and poor implant positioning (e.g., asymmetry due to lateral or cranial malposition or capsular contracture). In the literature, the main indication for pocket conversion is AD, which has been observed in more than 50% of patients who underwent submuscular implant-based BR, significantly worsening the aesthetic result. Lentz and Alcon [15] recently examined the impact of this complication on patient’s quality of life, reporting that about 80% of women were bothered by AD and 48% of women experienced an interference with their daily life activities. In our case series, we recorded animation deformity resolution in 100% of cases. This was in line with the findings by Sbitany et al. [6], who reported that pocket exchange corrected animation deformity in 100% of the cases in their 55 patient series, alongside an acceptable complication rate. They also observed that the application of pre-operative fat grafting contributed towards lower rates of complications and decreased the need for revisionary procedures. Only 16% of our patients (5 patients) underwent lipofilling before the procedure. It is interesting to note that these preliminary sessions of lipofilling were performed on the five patients who were submitted to radiotherapy before the pocket exchange, and they contributed to effectively enhancing the soft tissue thickness and quality. Our experience led us to the same conclusion as Sbitany et al., that performing lipofilling or a lipofilling series before the main operation provides thicker mastectomy flaps in cases of soft tissue insufficiency, and thus a better surgical field characterized by better quality tissue. Lipofilling enhanced the thickness of the soft tissue above the pectoralis major, allowing for faster and safer dissection, even in those patients with poor quality soft tissue or thinner mastectomy flaps. The pocket conversion procedure turned out to also be effective in the field of chronic pain; an immediate post-procedure resolution of this issue was observed in these patients. This is probably due to the reduction or elimination of the stimulation on some trigger points, such as intercostal lateral nerves, which are often involved in periprosthetic capsule formation, or can be associated with the detachment of the pectorals fibres to the implant capsule, lowering the chronic load of the pectoralis major, improving the overall motility of the shoulder joint. In our case series, a high grade of capsular contracture (Becker III/IV) was associated with waterfall deformity caused by the relative mastectomy flap’s ptosis on the contracted capsule, resulting in an unpleasant aesthetic outcome. We managed it with partial or sub-total capsulectomy, which was performed to increase the expansion of the skin envelope and obtain a more natural cosmetic result and a more natural fullness of the lower pole, consequently diminishing the upper pole fullness. There are different possibilities to manage the capsule; in our experience, they are dictated by the capsular contracture degree and by specific patient necessity. Patients with a lower grade of capsular contracture received only capsulotomies (12 breasts), both radial and circumferential, which let us to achieve a correct level of skin extension. Patients with a higher degree of capsular contracture associated with an implant cranialization were treated with subtotal capsulectomies, where subtotal capsulectomy means the removal of at least the caudal part of the anterior capsular layer or the whole anterior capsular layer if the upper part of the anterior layer of the capsule jeopardised the final result. In contrast with what Swanson E. stated regarding the treatment of capsular contracture in breast augmentation patients [16], we more frequently performed capsulectomies than capsulotomies, both for a conspicuous number of Becker III and IV capsular contractures (19 breasts), and to widely improve the expression of scarcely curved lower poles. Our experience dictates that, besides gentle surgical handling [17], one of the most important factors in pocket conversion is proper clinical evaluation of tissue thickness in all breast quadrants. We retrospectively determined that a post-mastectomy flap pinch test lower than 2 cm made the surgery more difficult, and exposed the patient to the risk of complications [18], such as tissue damage or wound dehiscence, and consequently implant exposure. Based on our, albeit limited, clinical series, we developed an algorithm (Figure 3) to highlight the correct steps for obtaining good outcomes.

The algorithm is mainly driven by clinical assessment (CC, CP, AD, implant malposition and pinch test, and distinguishing between upper and lower pole) and instrumental evidence (CC and potential implant rupture). If implant rupture is suspected (left-orange chart, Figure 3), capsulectomy should be sub-total, bearing in mind that capsule adherence to the ribs makes complete removal almost impossible, besides not being necessary. Thus, the skin flaps are assessed: the pre-operative pinch test should be greater than 2 cm and intraoperative evaluation should confirm this. Furthermore, flaps should be viable and reliable. Only under these conditions is pocket exchange indicated, otherwise it is not recommended. If during the clinical examination (right-blue chart, Figure 3) the pinch test is adequate (>2 cm) at both the upper and lower poles, we opt for simple removal of the implant, and capsulotomy when necessary. If the thickness of the upper or lower pole, or both, is poor (pinch test < 2 cm), this is an indication for lipofilling, with the patient to be clinically reassessed three months after to determine whether the thickness is sufficient for pocket exchange. Although it is too early for statistical evidence to demonstrate the validity of this algorithm, we are recording better results in-house, with lower complication rates, and this decision-making approach may thus be of interest to other institutions. Longer follow-up, larger patients’ recruitment, and accurate evaluation of the results are necessary to standardize this algorithm and to make it a valid and reproducible protocol, even outside the local reality.

## 5. Conclusions

The positive experiences with the pre-pectoral breast reconstruction have led to the will to resolve retro-pectoral reconstruction drawbacks such as animation deformity, which are not seen in the former. With these premises, the concept of the pocket conversion was introduced and the enthusiasm for this technique was confirmed by the positive outcome, such as 100% of animation deformity resolution. While different variants of this technique have been well described in the literature, no article has shown a flowchart on how to perform it in a safe way with the best tissue condition. We believe that our decisional algorithm could be a basic guide to reach the best and safest result. Further study could confirm our little experience or improve it.

## Figures and Tables

**Figure 1 healthcare-11-00671-f001:**
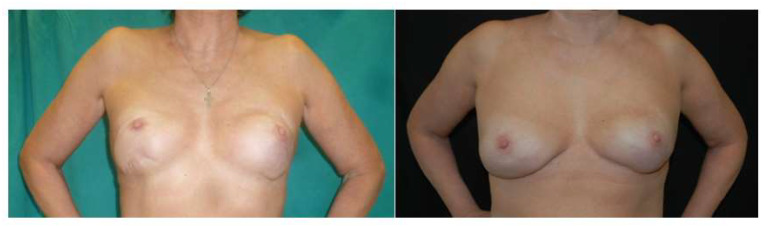
Bilateral implant animation deformity: pre-operative and 9-month post-operative view.

**Figure 2 healthcare-11-00671-f002:**
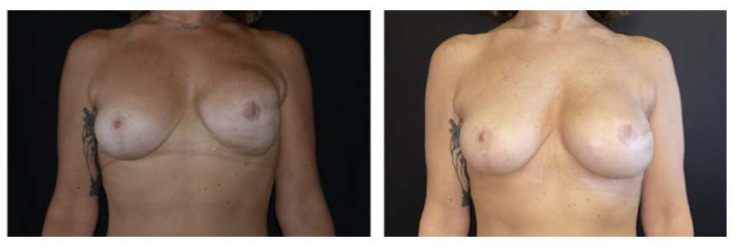
Pre-operative and 12-month post-operative view of implant pre-pectoral conversion of the left breast.

**Figure 3 healthcare-11-00671-f003:**
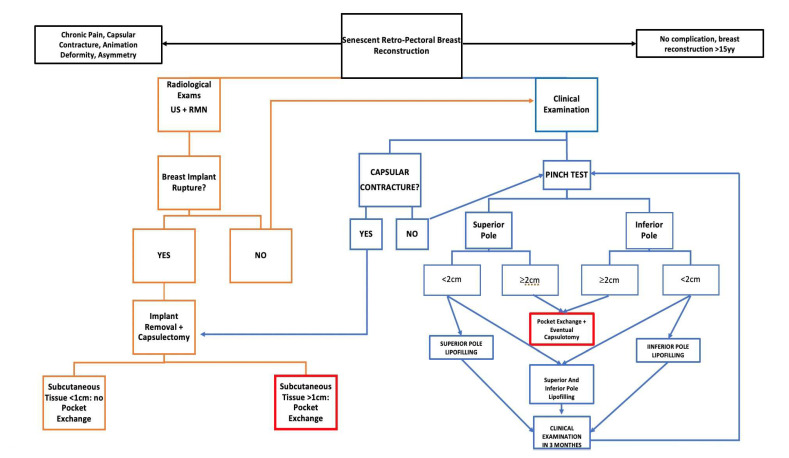
Decisional algorithm for breast pocket exchange.

**Table 1 healthcare-11-00671-t001:** Patients’ characteristics, capsular grade of contracture, and primary and secondary breast implant characteristics.

Patient characteristics	Mean BMI	24.9
	Mean Age	51 years
	Smoking	4/30 (13%)
	Hypertension	6/30 (20%)
	Diabetes	1/30 (3%)
	Previous breast radiotherapy	5/30 (17%)
Type of mastectomy	Nipple-sparing	16/31 (52%)
	Skin-sparing	12/31 (39%)
	Skin-reducing	3/31 (10%)
Previous implants characteristics	Volume 375 cc (mean volume)	
	Microtextured (anatomical)	29/31 (94%)
	Polyurethane (anatomical)	1/31 (3%)
	Smooth (round)	1/31 (3%)
Becker’s grade of capsular contracture	II	12/31 (39%)
	III	13/31 (42%)
	IV	6/31 (19%)
Secondary implants characteristics	Volume 395 cc (mean volume)	
	Microtextured (anatomical) - ADM coated - not ADM coated	28/31 (90%) 16/31 (52%) 12/31 (39%)
	Polyurethane (anatomical)	3/31 (10%)

## Data Availability

Not applicable.

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
