# Peer review of "From Sub-Pectoral to Pre-Pectoral Implant Reconstruction: A Decisional Algorithm to Optimise Outcomes of Breast Replacement Surgery"

_healthcare, 2023, doi:10.3390/healthcare11050671_

Round 1
Reviewer 1 Report
- Indication (type of mastectomy) is repeated in the abstract.
- There is no mention what type of study this is (eg. Retrospective chart review)
- Need to clarify if partial capsulectomy was done for becker 2 only? (line 82-83)
- Need better wording for what is done in the capsule in line 89-90.
- Lines 108-112 need better wording. Difficult to understand.
- It is assumed that all patients had unilateral procedures. I think mentioning it is worth stating even though it is obvious.
- Tables 1:
o Does average BMI refer to the mean BMI?
o Add percentages of demographics
o Previous implants characteristics:
§ Volume 375cc (on average): is this mean, median, or mode?
o Secondary implants characteristics:
§ Volume 395cc: Did all patients receive the same implant size?
- Lines 121-123:
o It is inferred that patients might have more than one indication for change of plane, but this was never stated on quantified. How many of the patients had more than one indication and how many had single indication for surgery?
- Did the mentioned complications occur in patients who were smokers, diabetic, hypertension, or had previous therapy? This information would make the study valuable. Also, despite a small sample size, statistical analysis can be done to compare demographics of patients who had complications vs those who did not. Why was age not mentioned as part of patient demographics?
- Were the patients that had lipofilling that same that had radiation?
Author Response
Thank you to the reviewer for her/his suggestions.
- we we have corrected the repetition reported in the abstract;
- we added the type of study it is also in the material and methods, actually, it was specified only in the abstract and not in the text;
- Partial capsulectomy was mainly performed in Becker 3 cases, and simple capsulotomies in Becker 2 (we specified that in the text)
- we better explained the surgery step by step both in line 89-80 and 108-112;
- thanks to the reviewer observation we realised that we wrongly reported our data. Actually, we have 30 patients and 31 breasts, it mean that we have a bilateral case, which is the one whose pictures are reported in Figure 1. We made the appropriate corrections in the abstract and in the text.
- I made all the suggested corrections at Table 1.
- The text is enriched with some quantitative information on the distribution of the indications for surgery as the reviewer requested.
- Complications occurred in patients who had 1 or more risks factors but as the reviewer has already highlighted it hasn't any statistical significance due to the small number of patients enrolled. However, we now specified in the text that all three patients who had complications had 1 or more risk factors. We hope in the future to collect a significative number of cases so that a statistical analysis could be done. We add the information about the mean age in demographics.
- The patients who had lipofilling are the same who had radiation therapy, but for the small number of cases any analysis has been performed. Nevertheless, I pointed out this detail in the text.
Reviewer 2 Report
The Authors present an interesting study on Sub-pectoral to pre-pectoral implant placement. I find the article well written and interesting for the general audience. I would recommend minor language editing such as in page 6 line 184 where the word Indeed should be replaced. Moreover the think the paper would benefit from the inclusion of pictures of a second case.
Author Response
Thank you for your positive feedback and suggestions. We will make the suggested changes and provide pictures of a second case.
Reviewer 3 Report
The authors present an interesting study on pocket conversion in implant-based breast reconstruction. I appreciated the decisional algorithm, which can be useful for young surgeons in their decision-making process.
I would recommend the authors to include more pictures of clinical cases (the intraoperative steps would be appreciated).
They should also explain how they assessed efficacy of fat grafting (clinical evaluation, ultrasound etc...) in the 5 patients who underwent lipofilling before pocket conversion. Can lipofilling be performed in the same operative time (with capsulotomy) and if not, why it is not recommended?
Author Response
Thank you for your considerations and suggestions.
We will provide pictures of a second clinical case; unfortunately we don’t have a worthy sequence of pictures of the intraoperative steps.
As we stated in the Manuscript, the preliminary sessions of fat grafting performed in the 5 patients were not initially carried out with a view to a future implant pocket conversion. The efficacy of fat grafting was assessed each time clinically, via pinch test and evaluation of the cutaneous envelope softness and quality. We prefer not to perform fat grafting at the same time as a prosthetic revision surgery in which capsulotomies have been performed. The interruption of the continuity of the prosthetic capsule in fact exposes to the risk of inoculating the adipose tissue inside the prosthetic pocket, which in turn exposes to the risk of infections and inflammatory reactions due to the not integration of the fat graft and the resulting liponecrosis.
Round 2
Reviewer 1 Report
Corrections are appreciate it. I believe the paper has improved merit in the current form.
Reviewer 2 Report
The paper can be accepted in the present form